# XMEA: A New Hybrid Diamond Multielectrode Array for the In Situ Assessment of the Radiation Dose Enhancement by Nanoparticles

**DOI:** 10.3390/s24082409

**Published:** 2024-04-10

**Authors:** Patricia Nicolucci, Guilherme Gambaro, Kyssylla Monnyelle Araujo Silva, Iara Souza Lima, Oswaldo Baffa, Alberto Pasquarelli

**Affiliations:** 1Department of Physics, Faculty of Philosophy, Sciences and Letters at Ribeirão Preto, University of São Paulo, Ribeirão Preto 14040-901, Brazil; nicol@usp.br (P.N.); iara.lima@usp.br (I.S.L.); baffa@usp.br (O.B.); 2Institute of Electron Devices and Circuits, University of Ulm, 89069 Ulm, Germany

**Keywords:** multielectrode array (MEA), diamond detector, nanoparticles, radiation dose enhancement

## Abstract

This work presents a novel multielectrode array (MEA) to quantitatively assess the dose enhancement factor (DEF) produced in a medium by embedded nanoparticles. The MEA has 16 nanocrystalline diamond electrodes (in a cell-culture well), and a single-crystal diamond divided into four quadrants for X-ray dosimetry. DEF was assessed in water solutions with up to a 1000 µg/mL concentration of silver, platinum, and gold nanoparticles. The X-ray detectors showed a linear response to radiation dose (r^2^ ≥ 0.9999). Overall, platinum and gold nanoparticles produced a dose enhancement in the medium (maximum of 1.9 and 3.1, respectively), while silver nanoparticles produced a shielding effect (maximum of 37%), lowering the dose in the medium. This work shows that the novel MEA can be a useful tool in the quantitative assessment of radiation dose enhancement due to nanoparticles. Together with its suitability for cells’ exocytosis studies, it proves to be a highly versatile device for several applications.

## 1. Introduction

The use of nanoparticles to increase radiation therapy efficacy has been studied for more than a decade, given its potential to increase the biological response to a radiation dose [1,2,3,4,5,6,7,8,9]. High-atomic-number nanoparticles embedded into the tumor increase its effective atomic number, enhancing the attenuation of the radiation and the dose to the biological medium. Additionally, nanoparticles have the potential to increase the biological tissue radiosensitivity. Monte Carlo simulation has been extensively used to study the interaction of the radiation with nanoparticles and the production and characteristics of the secondary particles produced. These particles, ejected from the nanoparticles to the medium, have lower energy and shorter range, creating a local effect of dose enhancement and radiosensitization [10,11,12,13,14,15]. Given the nano-to-micrometric range of the secondary charged particles liberated by the nanoparticles to the surrounding medium, experimental methodologies aiming to assess the dose enhancement are still challenging, since the dosimeter volume must be in the close vicinity of the nanoparticles. Among the dosimetry methodologies used to experimentally study the dose enhancement factor (DEF) produced by nanoparticles are electron spin resonance (ESR) [16,17], gel dosimetry [18,19,20], and chemiluminescence [21]. Even though the dosimeter volume is in the vicinity of the nanoparticle in those techniques, the medium is fairly different from the biological tissue where the dose enhancement should be measured. Hence, an experimental setup and methodology is still necessary for radiobiology and clinical studies of nanoparticle dose enhancement and radiosensitization.

Electrochemical biosensor arrays employing different materials and configurations and allowing chemical functionalization for different applications have been studied [22,23,24,25,26]. Also, diamond-doped electrodes as electrochemical sensors have been used for applications such as water treatment and the development of biosensors [27,28].

Due to its physical and chemical properties, diamond is considered an excellent material for the detection of high-energy photons, ranging from deep-UV to X- and gamma rays. Several groups worldwide developed diamond X-ray detectors and dosimeters, both on single crystals and on polycrystalline diamond [29,30,31,32,33,34,35,36]. Diamond detectors have been used to quantify the radiation dose in external beam radiation therapy using X-rays and particle beams [37,38,39]. In essence, diamond X-ray detectors work as photodiodes following the mechanism of the inner photoelectric effect. It means that incoming photons with energies larger than the diamond bandgap of 5.45 eV, equivalent to wavelengths shorter than 225 nm, can be absorbed and generate electron–hole pairs. Such photogenerated carriers are separated by the internal electric field in the junction region, so that electrons are collected at the cathode, while holes move to the anode. By operating the photodiode in short-circuit mode by means of a transimpedance amplifier, the measured photocurrent results are linearly proportional to the irradiance.

A diamond multielectrode array (MEA) device was originally designed to study the chemical products of exocytotic events from cells [40], but its unique architecture, combining the diamond electrodes to a cell-culture well, would also potentially allow the quantification of the radiation dose to the cell cultures, given the radiation sensitivity of the electrode material [41].

Aiming for the real-time detection of radiation-induced electrophysiologic and secretory activity of biological cells and tissues, and the quantification of the radiation dose, a novel hybrid diamond-MEA device, the XMEA, was developed. In this work, its suitability for assessing the radiation dose enhancement produced by metallic nanoparticles in aqueous media irradiated with low-energy X-rays is presented.

## 2. Materials and Methods

### 2.1. XMEA

The XMEA is a stack of two devices. The upper device is a planar 16 channel MEA for the detection of electrophysiological events, like exocytosis via amperometry or action potentials via potentiometry [40,41,42,43,44,45]. The basic fabrication of the nanocrystalline diamond (NCD) MEA is described in [45]. However, for this XMEA application, after finishing the NCD-MEA fabrication on a silicon wafer, the stack of intrinsic diamond, boron-doped diamond, and passivation were transferred from the silicon carrier to a glass plate, in order to combine low noise and high sensitivity with transparency. This transfer technique consists of the following steps: (1) glue the top side of the donor MEA onto a sapphire plate with an acetone-solvable glue, (2) dissolve the silicon oxide (SiO_2_) interlayer between the silicon carrier and the intrinsic diamond layer with hydrofluoric acid, (3) glue the back side of the intrinsic diamond layer to the final glass carrier with an acetone-resistant optical-grade glue, and (4) dissolve in acetone the first temporary glue, in order to remove the sapphire plate and clean the sensing top surface. Characterization of the MEA regarding the electrochemical detection is available in the Appendix A.

Figure 1 shows schematically the layout of the two devices and the overall assembly of the XMEA-stack. The lower device is based on a 10 × 10 × 1 mm^3^ heteroepitaxial single-crystal diamond grown using chemical vapor deposition [46], divided in four quadrants, each one structured as a photodiode (PD), for X-ray detection. Such photodiodes have an individual top layer consisting of a boron-doped diamond (BDD) layer 300 µm thick and a common back contact consisting of a 200 nm thick aluminum layer, which produces a Schottky contact. Therefore, the PD is a p-type/intrinsic/metal (p-i-m) stack. Thanks to the large carrier mobility, the high drift velocity and the lifetime of photogenerated e–h pairs in intrinsic diamond single crystals [47,48], the charge collection under exposure has a good yield even without biasing, i.e., by connecting the back contact to ground. However, it is possible to bias the back contact for increasing the internal electric field, thus enhancing the collection efficiency (e.g., with a bias of 8 V the signal amplitude is approximately doubled).

### 2.2. Nanoparticles

For the synthesis of silver, platinum, and gold nanoparticles, analytical-grade chemical reagents were used without modification or purification. All the syntheses were based on chemical reduction using a strong reducing agent and the metal precursor salt.

Silver nanoparticles (AgNPs) were produced by the reduction in an Ag(I) salt using an aqueous solution of silver nitrate (AgNO_3_) at a concentration of 2 mmol/L and a dilute solution of the reducing agent sodium borohydride (NaBH_4_, Sigma-Aldrich, Saint Louis, MI, USA) at a concentration of 4 mmol/L. The system was vigorously stirred overnight, resulting in colloidal dispersion with a dark-yellow color [49].

The synthesis of platinum nanoparticles (PtNPs) was carried out through the chemical reduction of chloroplatinic acid hexahydrate (H_2_PtCl_6_•6H_2_O, 99.999%, Sigma-Aldrich, Saint Louis, MI, USA) using an aqueous solution (2 mmol/L) of H_2_PtCl_6_•6H_2_O, which was added to a freshly prepared aqueous solution (4 mmol/L) of NaBH_4_. The system was vigorously stirred for 12 h to ensure complete reduction of the platinum ion. To prevent aggregation, 100 μL of 0.005 gm/L polyvinyl alcohol (PVA) aqueous solution was added to the mixture, resulting in a colloidal dispersion with a light-brown color.

Gold nanoparticles (AuNPs) were synthesized by reducing the gold salt Au (III) using an aqueous solution of chloroauric acid (HAuCl_4_, 99.999%, Sigma-Aldrich, Saint Louis, MI, USA) at a concentration of 2 mmol/L and a solution of NaBH_4_ at a concentration of 8 mmol/L. The system was vigorously stirred for 12 h at room temperature, resulting in a colloidal dispersion with a red color [50]. The stability of nanoparticle solutions over time was carefully monitored using characterization techniques. Our observations indicated no signs of nanoparticle sedimentation or aggregation. It is worth noting that we were meticulous in ensuring the use of fresh colloidal nanoparticle solutions in all of our analyses. This approach minimizes the possibility of external factors, such as changes in pH, temperature fluctuations, exposure to light, and contamination by foreign particles or ions, influencing stability assessments over time. Furthermore, variations in storage conditions, such as inadequate sealing of containers or exposure to air, can also affect the stability of nanoparticles.

UV–vis spectroscopy is a commonly used technique for characterizing noble metal nanoparticles, as it provides information about the size and shape of the particles. The absorption spectra of these nanoparticles are directly related to the surface plasmon resonance (SPR) phenomenon, which is strongly influenced by the size, shape, and composition of the particles [51,52]. Hence, the formation of nanoparticles was investigated by the UV–vis spectra using an Ultrospec 2100pro spectrophotometer (Amersham Pharmacia, Buckinghamshire, UK). The particle size distribution was acquired using the dynamic light-scattering (DLS) technique, using the Zeta-Sizer system (Malvern Instruments, Worcestershire, UK), at a fixed wavelength and angle (He–Ne 633 nm, 90° laser). The measurements were conducted in aqueous solvent at an ambient temperature, and the data acquisition parameters were standardized using the instrument’s software (Zetasizer Nano version 7.02). Subsequently, the obtained DLS data were processed and plotted using the Origin software version 8.0 for visualization and further analysis. A JEOL-JEM-100 CXI transmission electron microscope (TEM) (JEOL USA, Inc., Peabody, MA, USA) was used to analyze the morphology of the nanoparticles. For sample preparation, the nanoparticles were drop-cast onto a copper substrate coated with a conductive polymer and then dried at room temperature. The distribution of nanoparticle sizes was assessed using ImageJ software version 154i for particle counting, while Origin software was employed to derive the mean size through Gaussian fitting.

Table 1 shows the concentration for the aqueous solutions of AgNPs, PtNPs, and AuNPs used in the dose-enhancement assessments with the XMEA.

### 2.3. Irradiations

All irradiations were performed using an X-ray tube Konica Minolta (Tokyo, Japan), model Altus ST 543 HF. To check the XMEA detectors’ response linearity with X-rays, radiation beams were produced with exposure times varying from 32 ms up to 320 ms, with a tube current of 250 mA, and tube voltages of 50 kVp, 80 kVp, and 100 kVp. No added filtration was used and the corresponding half-value layers of 1.93 mmAl, 2.98 mmAl, and 4.13 mmAl, respectively, were achieved for the produced beams. A focal spot-to-sample detector was kept at 20 cm for all irradiations. A calibrated ionization chamber (PTW, model TN34069-2.5 with electrometer PTW Unidos E) was used to assess the air-kerma at each exposure.

The 100 kVp X-rays were employed for the dose enhancement measurements since that beam presents higher fluence at energies closer to the absorption energies of the nanoparticles than the lower energy X-ray beams.

### 2.4. Dose Enhancement Factor (DEF) and Response Enhancement Factor (REF)

For all measurements of DEF, 1.5 mL of aqueous solution of nanoparticles was used in the XMEA’s well, resulting in a 2 mm height solution in the well. The DEF produced in water by the nanoparticles was calculated as the ratio of the response of the XMEA diamond electrodes when the nanoparticle solution is present in the well and the response when only water is in the well (zero-nanoparticle concentration). The response of each XMEA electrode/channel is given by the integral under the signal (current) over the exposure time. The DEFs were calculated for each channel in contact with the aqueous solution of nanoparticles, and then averaged for each material and concentration. All measurements were made in triplicate.

Since the four detectors (photodiodes) of the XMEA are not in contact with the medium with nanoparticles, their signals were not used to assess DEF in the medium with nanoparticles. However, since secondary low-energy photons can also be ejected from the nanoparticles and reach these detectors, a response enhancement factor (REF), similar to the DEF, was calculated using the signals of these four detectors.

## 3. Results

### 3.1. Characterization of the Nanoparticles

Figure 2 shows the absorbance spectra of the nanoparticles, confirming its formation based on the plasmonic band of each material. According to the Mie theory, metallic nanoparticles with sizes smaller than the incident light wavelength exhibit a strong SPR absorption band in the visible to near-infrared (NIR) region, arising from the collective oscillation of free electrons on the surface of the nanoparticle [53].

The morphology and size distribution of the nanoparticles were initially examined using transmission electron microscopy (TEM), as shown in Figure 3. TEM analysis confirmed that all nanoparticles exhibit a spherical morphology with average sizes of 5.9 nm ± 0.09 for AgNPs, 3.5 nm ± 0.7 for PtNPs, and 4.4 nm ± 0.08 for AuNPs. It is worth noting that TEM analysis revealed a relatively narrow size distribution, indicating the predominance of similar sizes with little variation. Subsequently, dynamic light scattering (DLS) was employed to further assess the size distribution, as depicted in Figure 2d. DLS is valuable for monitoring stability, particularly concerning aggregation phenomena. Analysis via DLS revealed approximate sizes of 4.0 nm for AgNPs, 3.5 nm to 8.0 nm for PtNPs, and 7.0 nm ± 0.08 for AuNPs. Notably, only a single peak was detected for the nanoparticles, suggesting a relatively uniform size distribution and good stability.

However, it is crucial to recognize the disparity between the size distributions obtained from DLS and TEM analyses. Although TEM indicates smaller average sizes compared to DLS values, this incongruity may point to a polydispersity signal with a narrow distribution of nanoparticles. This implies that although average sizes may differ between the two techniques, there is still a predominance of similar sizes with little variation, indicative of a relatively narrow distribution of nanoparticles despite potential polydispersity. Therefore, although the nanoparticles may exhibit some polydispersity, they appear to maintain stability, as evidenced by TEM and DLS analyses with a narrow size distribution.

### 3.2. Response of the XMEA Detectors to X-rays

Figure 4 shows the typical response of the XMEA detectors when irradiated to X-rays produced with different exposure times and tube currents. The signal shown for each exposure is the averaged signal of the four X-ray detectors.

Since the signal of the detectors are given as the collected current over exposure time, the response of the detector for a given exposure is the area under the signal’s curve, representing the total charge collected by the detector.

Figure 5 shows the response of the XMEA detectors as a function of the air-kerma, measured with the calibrated ionization chamber. As can be seen, the response of the XMEA is linear for all the tested radiation qualities (r^2^ ≥ 0.9999).

Table 2 shows the sensitivities of the XMEA for the X-ray beams used, given by the slope of the linear fittings obtained. As expected, a higher sensitivity is found for a lower quality X-ray beam, representing a higher interaction of the radiation with the medium and the detector.

### 3.3. DEF

Figure 6 shows the DEFs obtained for the 100 kVp X-ray beam at various concentrations of AgNPs, PtNPs, and AuNPs. The DEFs for the PtNPs and AuNPs are larger than one, showing that these nanoparticles produced an increased dose to the surrounding medium. The AuNPs produced increasing DEFs with a nanoparticle concentration of up to 100 μg/mL, where the XMEA response saturated. Even though the atomic number of platinum and gold are similar, the produced DEFs are lower for the PtNPs, due to the partial attenuation of the produced secondary particles in its PVA outer layer. The AgNPs produced DEFs smaller than one, showing that these nanoparticles produced a shielding effect for the medium when irradiated with X-rays generated at 100 kVp, even for low concentrations.

Table 3 shows the DEFs assessed using the XMEA’s diamond electrodes for each nanoparticle material and concentration.

Figure 7 shows the REF for the X-ray detectors in XMEA for the different nanoparticles as a function of concentration. In a comparison with Figure 6, it can be seen that values of REF are overall lower than values of DEF for the PtNPs and AuNPs. For the PtNPs, intermediary concentrations even reduce the response of the detectors, due to the combination of an increased attenuation of the primary beam by the nanoparticles, preventing this radiation from reaching the detector, and the attenuation of the secondary radiation by the PVA coating. However, as the concentration of these two materials increases, the increased attenuation of the primary beam is compensated by the production of an increased fluence of secondary photons that reach the detectors, augmenting their response. The REFs produced by the AgNPs are higher than their corresponding DEFs since the shielding effect by these nanoparticles in the medium results in more low-energy photons reaching the four detectors.

## 4. Discussion

In this work, a novel diamond MEA device was employed to experimentally assess the dose enhancement produced by nanoparticles in the surrounding medium. Given the distinctive architecture of its diamond electrodes, positioned inside the device’s cell-culture well, it was possible to assess the DEF to water when different materials and concentrations of nanoparticles were used. Moreover, the diamond X-ray detectors of the XMEA allow for the coupled radiation dose monitoring during exposure. Even though the response of the device to radiation was obtained as a function of air-kerma (Figure 5), given the low-energy beam employed, it should be noted that the radiation dose is proportional to kerma, and a conversion coefficient can be used to calibrate the device’s response as a function of absolute dose values. Also, in the DEF calculations, the same conversion coefficients between the two dosimetric quantities (kerma and dose) will be used for the media with and without the nanoparticles, cancelling out in the ratio of the doses.

Even though more tests must be performed to fully characterize the detectors’ response to radiation, employing different beam qualities, radiation types, and dose rates, for example, the response obtained for low-energy X-rays shows the potential of the device. The linear response obtained for different X-ray doses indicates a suitable feature for a dosimeter. Also, the REF obtained for the X-ray detectors shows the potential to use those detectors to further explore the production of secondary photons by the nanoparticles.

Studies show that DEFs are relatively low when clinical radiation therapy beams are used, even for high-atomic-number nanoparticles [1,54]. The DEF is mainly due to secondary photoelectrons and Auger electrons produced in the nanoparticle, but the typical MeV primary beams employed in radiation therapy interact mainly via Compton scattering, hence producing a lower DEF than radiation beams in the keV range. The attenuation probability, and the main attenuation effect, will also be influenced by the nanoparticle’s material (atomic number).

For the nanoparticles used in this work, the higher DEFs were found for the AuNPs, as expected, since those have the highest atomic number. Even though the PtNPs have a similar atomic number, the PVA outer layer of those nanoparticles likely attenuates the low-energy secondary electrons, preventing them from depositing most of their energy in the medium. The smaller surface of the PtNPs also plays a role in the lower DEFs found, since the secondary electrons produced superficially in the nanoparticles are more likely to escape and increase the dose to the medium.

The results also demonstrated that the AgNPs produced a shielding effect for the medium. Different contributions can be mentioned to explain this behavior, the most important being the very low absorption energy of silver (K-edge at 25.5 keV, and L-edges at 3.8 keV, 3.5 keV, and 3.4 keV), and the greater reactivity of silver in relation to gold and platinum. Bhattarai et al. [55] have shown that silver is more reactive due to its lower ionization energy and electron affinity, which facilitates the removal of electrons from the outermost layer and leading to oxidation. This high reactivity can result in lower colloidal stability, increasing the likelihood of aggregation and auto-absorption of the produced secondary radiation.

As seen, the AuNPs showed the highest DEFs, given the higher atomic number of gold, leading to larger attenuation of the primary beam, and the higher absorption energy for production of secondary photoelectrons (K-edge at 80.7 keV, and L-edges at 14.4 keV, 13.7 keV and 11.9 keV) and Auger electrons. Given the small size of the AuNPs and the relatively low concentrations used in this work, the DEFs found for the AuNPs are relatively high in comparison to the literature, even considering a low-energy beam [1]. This can suggest that the DEFs may be higher than those calculated using Monte Carlo or methods where the experimental setup is not appropriate, as for example when large-volume dosimeters are employed. The DEF calculated with Monte Carlo is affected by the microscopic modelling of the nanoparticles, as well as the simulation parameters controlling the transport algorithm. Experimental studies can suffer from a lack of spatial resolution to measure nano and micro-scaled DEFs. Wolfe et al. used electron spin resonance (ESR) dosimetry to assess the DEF of gold nanoparticles mixed with 2-methyl-alanine and irradiated to up to 5 Gy with 250 kVp and 6 MV X-rays, showing the suitability of the technique [17]. However, Lima et al. showed complications of the technique depending on the medium used to incorporate the nanoparticles due to nanoparticle aggregation and radical recombination [16]. Nezhad et al. have tested MAGAT and PAG polymer gels to assess DEFs produced by bismuth oxide nanoparticles but concluded that interaction between the nanoparticles and the polymer monomers may prevent us from obtaining the expected results with this dosimetry technique [18].

Unfortunately, for the measurements with the AuNPs, the saturation of the electrodes signals at an intermediary concentration prevented further investigation of the DEFs in the medium surrounding the nanoparticles. Lowering the sensitivity of the electrodes (possible in the manufacture process) would allow the quantification of DEFs for higher AuNPs concentrations and could help in understanding the variation of DEF with concentration, since high concentrations tend to produce a shielding effect, lowering the dose to the medium as opposed to enhancing it. However, since the XMEA can also be used to quantify chemical composites release by cells during irradiation, which can happen at very low concentrations, the sensitivity of the electrodes should be chosen according to the desired application.

Another important aspect related to the nanoparticle concentration is its biological expected clearance and toxicity. Ganjeh and Salehi obtained a DEF of 1.8 when 10% concentration of PtNPs were incorporated in water and irradiated with protons of 2–15 MeV [53]. In this study, relevant DEFs were found for AuNP, even for the lower concentrations used. This was possible due to the novel diamond-MEA used in this work, which allowed the assessment of the DEF in situ in the relevant medium with nanoparticles incorporated. Moreover, the XMEA could be used to synchronously assess DEF, the physiological events of cellular exocytosis induced by radiation, and dose measurement, presenting excellent potential to allow true nanoparticle-mediated radiosensitization studies.

## 5. Conclusions

In this work, a diamond microelectrode array, originally designed for cells’ exocytosis studies, was employed to experimentally assess the radiation dose enhancement produced by nanoparticles. Comparisons for silver, platinum, and gold nanoparticles, irradiated with different X-ray beam qualities, showed that the combination of a high-atomic-number nanoparticle and a low-energy beam produced significant dose enhancement even when a low concentration of nanoparticle was used. Gold and platinum nanoparticles enhanced the dose to the surrounding medium, with the first producing the higher DEFs. Silver nanoparticles, on the other hand, produced a shielding effect for the medium in all the concentrations studied. The XMEA device was shown as an adequate system to assess dose enhancement factors, proving itself a highly versatile device, and paving the way to in vitro experiments on dose enhancement factors in cells.

## Figures and Tables

**Figure 1 sensors-24-02409-f001:**
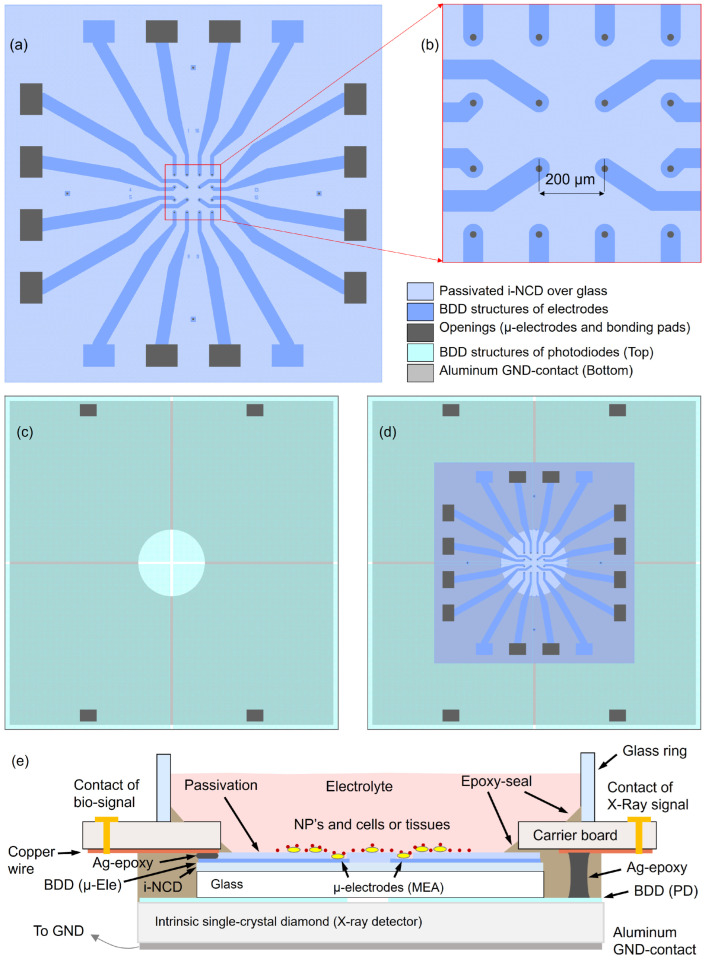
Structures and assembly of the XMEA: (**a**) Layout of the 16 Ch MEA. Channels 2, 7, 10, and 15 are not connected, leaving the connections available for the X-ray photodiodes; (**b**) magnification of the sensing area with 20 µm large µ-electrodes; (**c**) layout of the 4-quadrant X-ray detector consisting of p-i-m-photodiodes; (**d**) top view of the XMEA stack. The back aluminum contact has an opening for allowing the inspection of the MEA sensing area with an inverted microscope, thanks to the semi-transparency of the XMEA; (**e**) cross-section of the fully assembled XMEA. The glue between the intrinsic diamond layer (i-NCD) and the glass carrier is not shown.

**Figure 2 sensors-24-02409-f002:**
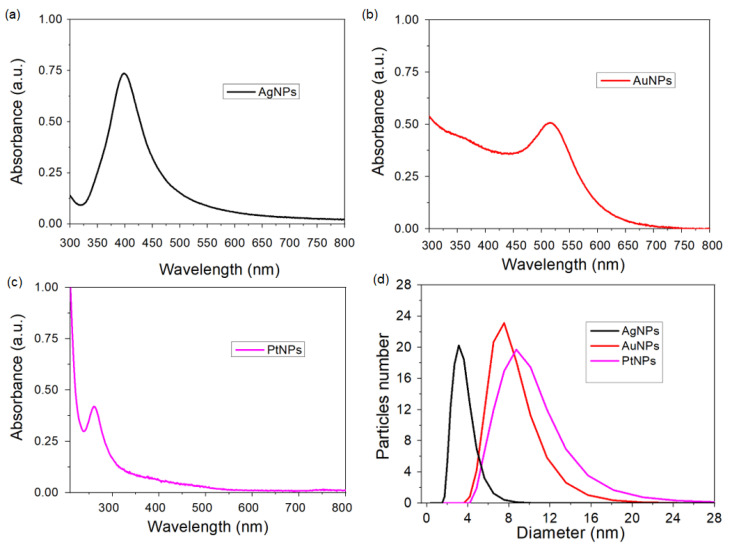
UV–vis spectra of colloidal solutions of different nanoparticles: (**a**) AgNPs, (**b**) AuNPs, and (**c**) PtNPs. The UV–vis spectra provide essential optical information about the nanoparticles, revealing characteristic absorption peaks that correspond to their unique electronic transition. (**d**) The average hydrodynamic size distribution of the metallic nanoparticles obtained using the dynamic light-scattering technique.

**Figure 3 sensors-24-02409-f003:**
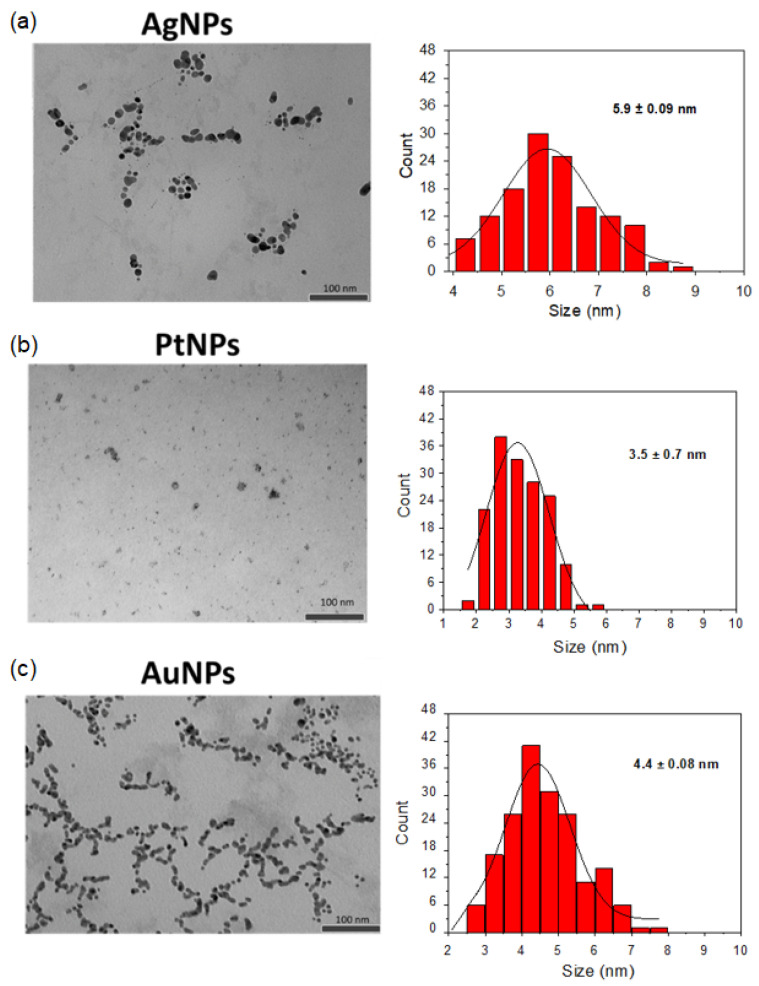
Transmission electron microscopy images (**left**) and size (diameter) distribution (**right**) of (**a**) AgNPs, (**b**) PtNPs, and (**c**) AuNPs. The images provide detailed visualizations of the spherical morphology and structure of each nanoparticle type, highlighting their distinctive characteristics.

**Figure 4 sensors-24-02409-f004:**
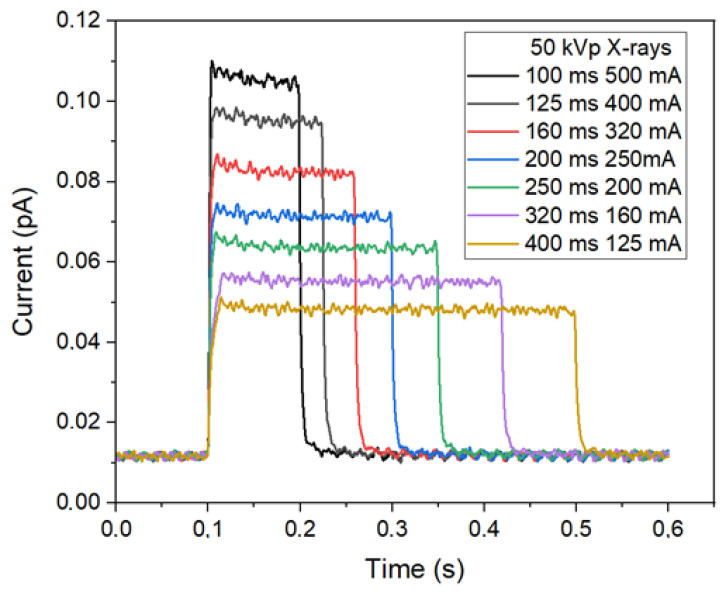
Typical signal of the radiation detectors of the d-MEA device when exposed to X-rays.

**Figure 5 sensors-24-02409-f005:**
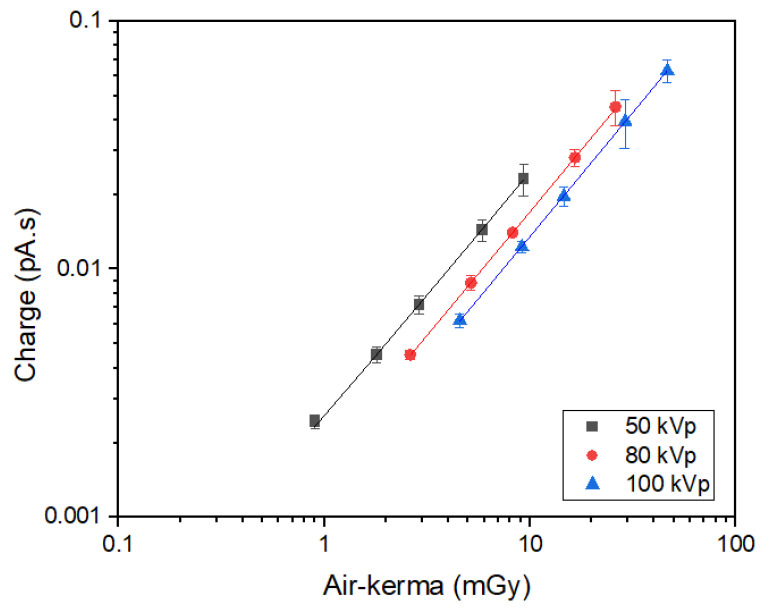
Response of the d-MEA as a function of air-kerma produced by X-rays.

**Figure 6 sensors-24-02409-f006:**
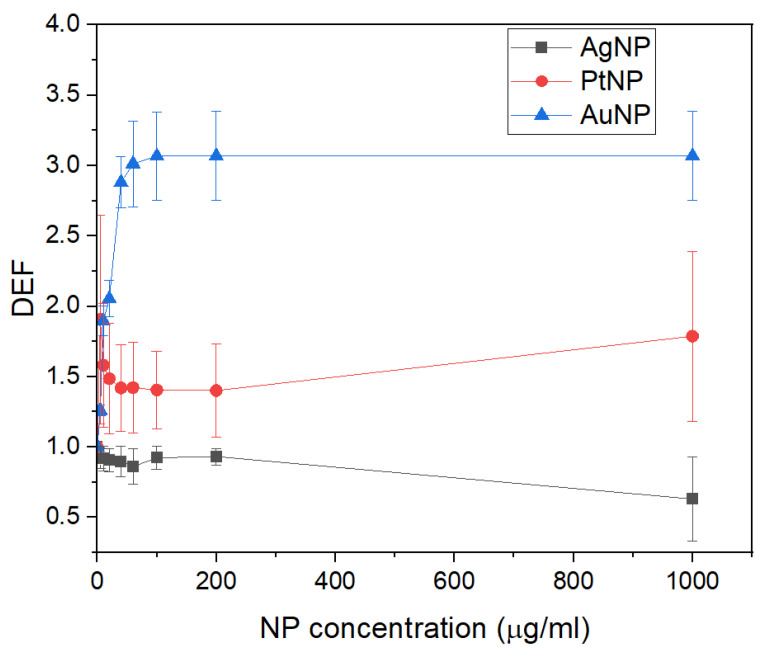
Dose enhancement factors obtained for the d-MEA device for AgNP, PtNP, and AuNP as functions of nanoparticle concentration.

**Figure 7 sensors-24-02409-f007:**
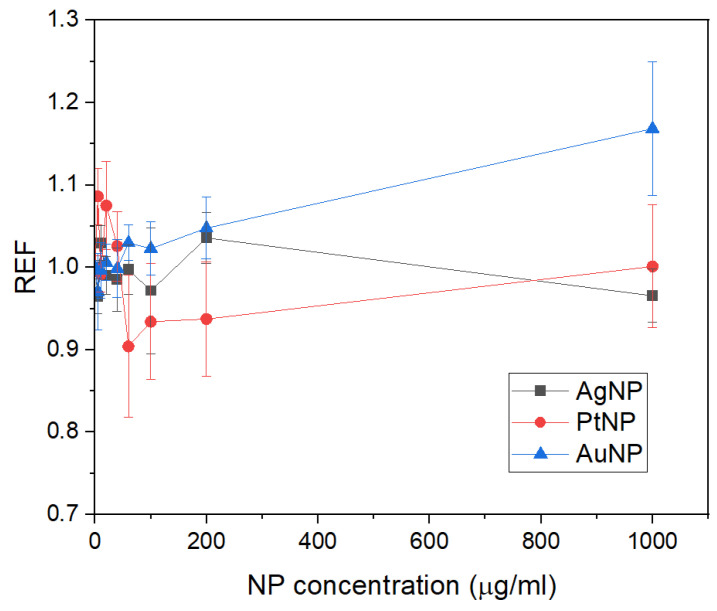
Response enhancement factors (REF) for the X-ray detectors of XMEA for AgNP, PtNP, and AuNP as a function of nanoparticle concentration.

**Table 1 sensors-24-02409-t001:** Concentration of the nanoparticles’ aqueous solutions.

Material	Concentration (µg/mL)	Molar Concentration (mol/mL)	Mass Percentage (% *w*/*w*)
Ag	5	4.63 × 10^−8^	0.0005
	10	9.27 × 10^−8^	0.001
	20	1.85 × 10^−7^	0.002
	40	3.71 × 10^−7^	0.004
	60	5.56 × 10^−7^	0.006
	100	9.27 × 10^−7^	0.01
	200	1.85 × 10^−6^	0.02
	1000	9.27 × 10^−6^	0.1
Pt	5	2.56 × 10^−8^	0.0005
	10	5.13 × 10^−8^	0.001
	20	1.03 × 10^−7^	0.002
	40	2.05 × 10^−7^	0.004
	60	3.08 × 10^−7^	0.006
	100	5.13 × 10^−7^	0.01
	200	1.03 × 10^−6^	0.02
	1000	5.13 × 10^−6^	0.1
Au	5	2.54 × 10^−8^	0.0005
	10	5.08 × 10^−8^	0.001
	20	1.02 × 10^−7^	0.002
	40	2.03 × 10^−7^	0.004
	60	3.05 × 10^−7^	0.006
	100	5.08 × 10^−7^	0.01
	200	1.02 × 10^−6^	0.02
	1000	5.08 × 10^−6^	0.1

**Table 2 sensors-24-02409-t002:** Sensitivity of the d-MEA device depending on the X-ray tube potential.

	50 kVp	80 kVp	100 kVp
Sensitivity (pA/mGy) ×10^−4^			
	24.6 ± 0.14	17.0 ± 0.09	13.5 ± 0.03

**Table 3 sensors-24-02409-t003:** DEFs for the AgNP, PtNP, and AuNP as functions of nanoparticle concentration.

Material	Concentration (µg/mL)	DEF
Ag	5	0.919 ± 0.074
	10	0.918 ± 0.086
	20	0.906 ± 0.082
	40	0.897 ± 0.108
	60	0.862 ± 0.126
	100	0.923 ± 0.081
	200	0.932 ± 0.058
	1000	0.631 ± 0.297
Pt	5	1.907 ± 0.741
	10	1.518 ± 0.440
	20	1.486 ± 0.393
	40	1.419 ± 0.306
	60	1.423 ± 0.320
	100	1.406 ± 0.275
	200	1.401 ± 0.330
	1000	1.788 ± 0.605
Au	5	1.256 ± 0.039
	10	1.896 ± 0.103
	20	2.054 ± 0.129
	40	2.881 ± 0.183
	60	3.013 ± 0.304
	100	3.067 ± 0.315
	200	3.068 ± 0.316
	1000	3.068 ± 0.316

## Data Availability

The raw data supporting the conclusions of this article will be made available by the authors on request.

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
