# Peer review of "XMEA: A New Hybrid Diamond Multielectrode Array for the In Situ Assessment of the Radiation Dose Enhancement by Nanoparticles"

_sensors, 2024, doi:10.3390/s24082409_

Round 1

Reviewer 1 Report

Comments and Suggestions for Authors

Before evaluating a text, it is necessary to obtain clear and precise answers from the authors, which relate to the following aspects:

1. How is the absorbed dose in the sample related to the MEA reading?

2. What is the emission spectrum and how does it affect the readings? Is the device response linear as a function of photon energy?

3. When passing through a medium, especially one containing elements with high atomic number, low-energy photon radiation is greatly attenuated. How is the decrease in photon flux and the increase in dose separated?

4. It is indirectly assumed that the increase in the absorbed dose is uniform over the volume and does not depend on the direction of the radiation. This must either be justified or the amendments taken into account using calculation methods.

In addition, it is highly desirable to see analytical estimates of DEF for given nanoparticles and photon spectra.

Author Response

Before evaluating a text, it is necessary to obtain clear and precise answers from the authors, which relate to the following aspects:

  1. How is the absorbed dose in the sample related to the MEA reading?

The radiation beams employed are calibrated in terms of air-kerma with a calibrated dosimeter (PTW, model TN34069-2.5, ionization chamber with electrometer PTW Unidos E, as described in section 2.3). Since the goal was assessing the dose enhancement factor, and not absolute dose to the sample at each exposure, using the air-kerma, that is proportional to the dose, is adequate. Hence, Figure 5 shows the “dose-response” curve of the MEA device. In that figure, the reading is given as the charge collected by the electrodes (integral of the collected current over the exposure time – Fig. 4) as a function of the air-kerma for each radiation beam.

  1. What is the emission spectrum and how does it affect the readings? Is the device response linear as a function of photon energy?

The X-ray spectra are specified by the half-value layers, as described in section 2.3. The energy dependence can be inferred from Fig. 5: different spectra result in linear fittings with different slopes, characterizing the energy dependence of the device. The slope for each linear fitting is presented in Table 2.

However, it’s worth noticing that all DEF data presented was obtained for the 100 kVp beam.

  1. When passing through a medium, especially one containing elements with high atomic number, low-energy photon radiation is greatly attenuated. How is the decrease in photon flux and the increase in dose separated?

In principle, those components can be separated using Monte Carlo simulation (see, for example, reference 11). Monte Carlo simulation studies show that, at low nanoparticle concentrations, there is not a significantly dose enhancement due to change in the secondary photon flux. The increase in dose is mainly due to the short-range-electrons around the nanoparticles.

In experimental approaches, the sensitive volume of the dosimeter must be in the close vicinity of the nanoparticles to be able to take that secondary electrons’ dose component into consideration. The proposed use of the XMEA lies exactly in considering that dose component in a short range from the nanoparticles, since the device’s micro-electrodes are in contact with the medium where the nanoparticles are dispersed.

  1. It is indirectly assumed that the increase in the absorbed dose is uniform over the volume and does not depend on the direction of the radiation. This must either be justified or the amendments taken into account using calculation methods.

The use of the XMEA to assess dose enhancement does not assumed that the increase in dose is uniform over the volume. Actually, the increase in dose to water surrounding nanoparticles far away from an electrode may not increase its collected charge. Here lies the greatest potential of using this device in radiobiology assays: since cells (with nanoparticles incorporated) can be inoculated in the vicinity of the electrodes, the quantified dose enhancement will be related to the dose enhancement to the cells, and not to the entire medium.

In addition, it is highly desirable to see analytical estimates of DEF for given nanoparticles and photon spectra.

Several aspects make it difficult to implement an accurate model to analytically estimate DEF: the heterogeneous distribution of the nanoparticles in biological media; its aggregation inside cells; the short range of the secondary electrons; the influence of nanoparticles concentration (for high concentrations, the nanoparticles can absorb the produced secondary electrons, actually decreasing the dose to the medium). Hence, Monte Carlo simulation studies are taken as the gold standard for DEF assessment, instead of analytical calculation. However, it should be noted that even Monte Carlo studies are highly affected by the geometry model employed for the nanoparticles dispersion in cell media.

Acknowledging all these factors, the scientific community has been trying to propose an experimental approach to DEF assessment. In that sense, this work’s goal is to explore a new alternative for that, instead of proposing an analytical approach to it.

Reviewer 2 Report

Comments and Suggestions for Authors

The manuscript entitled " XMEA: a new hybrid diamond multielectrode array for the in situ assessment of the radiation dose enhancement by nanopar- 3 ticles" has been reviewed according to the invitation by the Sensors editorial team. The manuscript discussed the fabrication and characterization of the multielectrode array for dose enhancement assessment. The following comments for authors were provided.

  1. The title must be free of abbreviation.
  2. Few writing errors are observed in the manuscript. The whole and proper form of chemical formula must be written, and respect the subscript.
  3. In the abstract, a few important results must be provided for readers.
  4. In the introduction, the physics of nanodiamond performance in detecting high-energy photons should be briefly mentioned for the readers.
  5. Nanoparticles prepared from sodium borohydride or sodium citrate do not have any steric stability like platinum nanoparticles that benefit from PVA coating. In previous studies, it has been shown that complex particles of small gold nanoparticles of 2 nm in liposomes absorb more radiation (https://doi.org/10.1007/s12032-023-01991-1). In addition to the difference in surface chemistry and stability, another issue is the difference in the surface charge of nanoparticles. Is this addressed in the evaluations? The authors should be able to justify why they did not use all three nanoparticles with the same physicochemical properties.
  6. Nanoparticles without proper coating are easily agarized in culture medium, and the pattern of radiation absorption in agar particles is different from dispersed particles.
  7. The difference between hydrodynamic diameter and TEM diameter must be discussed.
  8. Figure 2d represents the hydrodynamic diameter of the particles. "a. u." is not a proper unit for particle number. Also, more details are necessary for the provided diagram. 
  9. Considering that the XMEA electrodes have already been developed, the authors have not provided any characterization of them in the present manuscript. But if there are characterizations of these electrodes, it is better to provide them in this manuscript for the readers.
Comments on the Quality of English Language
  1. Few writing errors are observed in the manuscript. The whole and proper form of chemical formula must be written, and respect the subscript.

Author Response

The manuscript entitled " XMEA: a new hybrid diamond multielectrode array for the in situ assessment of the radiation dose enhancement by nanoparticles" has been reviewed according to the invitation by the Sensors editorial team. The manuscript discussed the fabrication and characterization of the multielectrode array for dose enhancement assessment. The following comments for authors were provided.

1) The title must be free of abbreviation.

If you mean the acronym “XMEA”, we are sorry but we do not agree with your suggestion. This acronym gives the title a more effective impact than a word-based description.

2) Few writing errors are observed in the manuscript. The whole and proper form of chemical formula must be written, and respect the subscript.

Writing errors have been hunted and corrected as much as possible; chemical formulas are now explicitly named at their first occurrence; subscripts and superscripts are implemented. The corrections are highlighted in the manuscript.

3) In the abstract, a few important results must be provided for readers.

Given the limitation on the number of words in the abstract, only some few important results were incorporated. Modifications are highlighted in the abstract.

4) In the introduction, the physics of nanodiamond performance in detecting high-energy photons should be briefly mentioned for the readers.

A description of the photodiode principle of operation is now available (highlighted in the text).

5) Nanoparticles prepared from sodium borohydride or sodium citrate do not have any steric stability like platinum nanoparticles that benefit from PVA coating. In previous studies, it has been shown that complex particles of small gold nanoparticles of 2 nm in liposomes absorb more radiation (https://doi.org/10.1007/s12032-023-01991-1). In addition to the difference in surface chemistry and stability, another issue is the difference in the surface charge of nanoparticles. Is this addressed in the evaluations? The authors should be able to justify why they did not use all three nanoparticles with the same physicochemical properties.

The nanoparticles were synthesized via chemical reduction using sodium borohydride. The stability of silver (Ag) and gold (Au) nanoparticles is attributed to the negative surface charge that surrounds the particles, guaranteed by sodium borohydride which, in addition to reducing, stabilizes, which is suitable for maintaining stability in our system. Regarding platinum nanoparticles (PtNPs), the main objective of using polyvinyl alcohol (PVA) was to improve the definition of the plasmonic band. The PVA coating facilitated the emergence of a well-defined plasmonic band centered at 260 nm in the absorption spectrum due to probably expanded coordination during the division of the Pt 5d orbital binding field, as these nanoparticles present two unstable absorption bands in the UV region, in around 215 nm and 260 nm. Additionally, we would like to inform you that the article relating to this synthesis has been submitted to a journal and we are awaiting comments from reviewers.

In our study, nanoparticles (Ag, Au) synthesized via chemical reduction exhibit inherent stability due to their negative zeta potential, making additional surface coatings unnecessary. For nanoparticles (Ag, Au), our research group has standardized a synthesis method that guarantees high stability and high negative zeta potential. As for PtNPs, the main objective of using PVA coating was to improve the definition of the absorption band.

Also, our study was specifically focused on exploring the potential benefits of nanoparticles with higher atomic numbers to enhance their interaction with ionizing radiation. Therefore, we deliberately selected nanoparticles based on their atomic properties rather than their physicochemical similarities. It's worth noting that our research group has previously demonstrated the effectiveness of nanoparticles (Ag, Au) in increasing radiation dose in alanine dosimeters.

6) Nanoparticles without proper coating are easily agarized in culture medium, and the pattern of radiation absorption in agar particles is different from dispersed particles.

In our preliminary tests, the system was not exposed to a solution containing culture medium, only the colloidal suspension of NPs. However, we appreciate your insight as it may provide valuable considerations for future work comparing coated and uncoated nanoparticles. Ensuring the stability and behavior of nanoparticles in the culture medium is indeed crucial for an accurate assessment, and we will take this into account in future experiments.

7) The difference between hydrodynamic diameter and TEM diameter must be discussed.

In our manuscript, we indeed discussed the difference between the hydrodynamic diameter obtained by dynamic light scattering (DLS) and the diameter observed by transmission electron microscopy (TEM). This discussion is highlighted in section 3.1. We recognize the disparity between the size distributions obtained from these two techniques and discuss how this incongruity may suggest a polydispersity signal with a narrow nanoparticle distribution. We emphasize the importance of considering this difference in our analysis and its implications for nanoparticle stability.

8) Figure 2d represents the hydrodynamic diameter of the particles. "a. u." is not a proper unit for particle number. Also, more details are necessary for the provided diagram.

The unit for particle number was removed in the figure. Also, more details on the measurements were added in section 2.2 (highlighted in the text).

9) Considering that the XMEA electrodes have already been developed, the authors have not provided any characterization of them in the present manuscript. But if there are characterizations of these electrodes, it is better to provide them in this manuscript for the readers.

This is now available as supplementary material.

Reviewer 3 Report

Comments and Suggestions for Authors

In this paper, the authors using a diamond microelectrode array to assess the radiation dose enhancement effect produced by nanoparticles. Overall, there are no major problems with the grammar or results of this paper. But there are still some minor issues that arise in the paper. If the following issues are well-addressed, this reviewer believes that the essential contribution of this paper is important for Sensors.

1.     In line 376-377, the authors described their experimental setup is not appropriate, so they should discuss further on the issue and their results in the paper are also discussed. A description should be given of when and why the experimental setup is not applicable.

2.     The conclusion is too short and should summarise the results of the experiments with the three nanoparticles (platinum,silver and gold nanoparticles), making comparisons and generalisations.

3.     The scope of application about XMEA should be added on in the discussion.

4.     Graphing should be standardised. Figures 4-7 should be plotted as standard as Figure 2.

Comments on the Quality of English Language

None

Author Response

In this paper, the authors using a diamond microelectrode array to assess the radiation dose enhancement effect produced by nanoparticles. Overall, there are no major problems with the grammar or results of this paper. But there are still some minor issues that arise in the paper. If the following issues are well-addressed, this reviewer believes that the essential contribution of this paper is important for Sensors.

  1. In line 376-377, the authors described their experimental setup is not appropriate, so they should discuss further on the issue and their results in the paper are also discussed. A description should be given of when and why the experimental setup is not applicable.

The referenced phrase in the manuscript states that the experimental setup of other studies may not be appropriate, not the one used in this study. Actually, the size and position of the electrodes of the XMEA in the medium is an improvement in the measurement setup for DEF assessment. An example of a potentially unsuitable measurement setup was added to the text (highlighted).

  1. The conclusion is too short and should summarise the results of the experiments with the three nanoparticles (platinum,silver and gold nanoparticles), making comparisons and generalisations.

Comparison on the nanoparticles results were added to the conclusions (highlighted).

  1. The scope of application about XMEA should be added on in the discussion.

Even though the XMEA device has a broader scope of applications that assessing DEF, the data shown in the manuscript is restricted to that application. Hence, we highlighted two already existing phrases in the Discussions section emphasizing that scope:

“In this work, a novel diamond MEA device was employed to experimentally assess the dose enhancement produced by nanoparticles in the surrounding medium.”

“Moreover, the XMEA could be used to synchronously assess DEF, the physiological events of cellular exocytosis induced by radiation, and dose measurement, presenting excellent potential to allow true nanoparticles-mediated radiosensitization studies.”

  1. Graphing should be standardised. Figures 4-7 should be plotted as standard as Figure 2.

The figures were modified accordingly.  

Reviewer 4 Report

Comments and Suggestions for Authors

The research article by P. Nicolucci et al. describes a novel diamond microelectrode array (MEA) for measuring the dose enhancement caused by metal nanoparticles in the surrounding medium. The authors characterize the device’s structure, its assembling as well as its suitability for evaluating the radiation dose enhancement by nanoparticles irradiated with low energy X-rays. The experimental procedures are systematic and supported by the statistics. Figures are clear and the results are interesting with a high potential of disruptive applications. However, some points need to be addressed while other should be further discussed before considering, from my side, the manuscript suitable for Sensors journal. Below, the list of points need to be addressed.

-At lines 46-47 authors state: “Electrochemical biosensor arrays of different materials and configurations have been studied for several applications”. For completeness, a brief introduction to MEA, their applications as well as recent customizations in terms of electrodes number and also the type of materials/functionalization, should be added in the introduction. You may critically use the following work - Langmuir 2016, 32, 25, 6319–6327 https://doi.org/10.1021/acs.langmuir.6b01352. This would increase the readability of the manuscript and make it more accessible to experts out-of-the field especially in terms of future developments.

-At lines 132-133 authors state: “This approach minimizes the possibility of external factors influencing stability assessments over time.” Please, add some examples of these external factors affecting the stability.

- Figure’s caption is not coherent with figure’s labels. Please, modify it.

- The nanoparticles average size from the TEM analysis is reported in Results section; please add to the main text how the distributions have been calculated.

- At lines 283-285 authors state: “Since the signal of the detectors are given as collected current over exposure time, the response of the detector for a given exposure is the area under the signal’s curve, representing the total charge collected by the detector.” Have these areas been measured? If yes, please also add this data, otherwise it would be better to state it explicitly (for example data not shown)

- At lines 396-399 authors state: “since the XMEA can also be used to quantify chemical composites release by cells during irradiation, which can happen at very low concentrations, the sensitivity of the electrodes should be chosen according to the desired application.” I personally consider this feature the main relevant for future developments; this aspect should be further emphasized, (mentioning it for example also in the introduction or even in the abstract) as it gives extremely high versatility to the device examined opening the way for specific experiments and application in life science context.

Author Response

The research article by P. Nicolucci et al. describes a novel diamond microelectrode array (MEA) for measuring the dose enhancement caused by metal nanoparticles in the surrounding medium. The authors characterize the device’s structure, its assembling as well as its suitability for evaluating the radiation dose enhancement by nanoparticles irradiated with low energy X-rays. The experimental procedures are systematic and supported by the statistics. Figures are clear and the results are interesting with a high potential of disruptive applications. However, some points need to be addressed while other should be further discussed before considering, from my side, the manuscript suitable for Sensors journal. Below, the list of points need to be addressed.

  • At lines 46-47 authors state: “Electrochemical biosensor arrays of different materials and configurations have been studied for several applications”. For completeness, a brief introduction to MEA, their applications as well as recent customizations in terms of electrodes number and also the type of materials/functionalization, should be added in the introduction. You may critically use the following work - Langmuir 2016, 32, 25, 6319–6327 https://doi.org/10.1021/acs.langmuir.6b01352. This would increase the readability of the manuscript and make it more accessible to experts out-of-the field especially in terms of future developments.

The text was rearranged to reference studies on different applications of MEA on that same phrase. Also, the suggested reference was added to the manuscript. However, since this manuscript is not about the construction characteristics of device’s electrodes, but it’s use in a novel application, for clarity, a description of recent customizations and developments of MEAs’ was not included – there are, however, references about those aspects in the listed references.

 The changes in the text are highlighted in the Introduction.

  • At lines 132-133 authors state: “This approach minimizes the possibility of external factors influencing stability assessments over time.” Please, add some examples of these external factors affecting the stability.

We incorporated such information in section 2.2 (highlighted in the text).

  • Figure’s caption is not coherent with figure’s labels. Please, modify it.

We corrected the figure’s captions (highlighted in the text).

  • The nanoparticles average size from the TEM analysis is reported in Results section; please add to the main text how the distributions have been calculated.

We have updated the main text to clarify how nanoparticle size distributions were calculated in the Materials and Methods (section 2.2). Specifically, we used ImageJ software to analyze the diameter of more than 200 particles, enabling the generation of a histogram to represent the size distribution. This approach ensured accurate and comprehensive characterization of nanoparticle sizes, as reported in the Materials and Methods.

Modifications to the text are highlighted.   

  • At lines 283-285 authors state: “Since the signal of the detectors are given as collected current over exposure time, the response of the detector for a given exposure is the area under the signal’s curve, representing the total charge collected by the detector.” Have these areas been measured? If yes, please also add this data, otherwise it would be better to state it explicitly (for example data not shown)

Yes, these areas were calculated for each exposure of the device.

The referenced phrase means to explain how the response of the device is calculated for a given exposure. In this way, Fig. 4 shows typical signals for single exposures, and the area under the curve is the “response” of the device for that exposure. Additionally, all experiments are performed in triplicate and the averaged value is taken as response for the experiment. For the experiments of response versus dose, for example, the data is shown in Figure 5.

  • At lines 396-399 authors state: “…since the XMEA can also be used to quantify chemical composites release by cells during irradiation, which can happen at very low concentrations, the sensitivity of the electrodes should be chosen according to the desired application.” I personally consider this feature the main relevant for future developments; this aspect should be further emphasized, (mentioning it for example also in the introduction or even in the abstract) as it gives extremely high versatility to the device examined opening the way for specific experiments and application in life science context.

This was briefly mentioned in the abstract (due to the limitation of the allowed number of words) and in the conclusions. However, since the presented data is only related to the DEF assessment, that application remains emphasized. Modifications to the text are highlighted.   

Round 2

Reviewer 1 Report

Comments and Suggestions for Authors

There are many comments on the work, but I see no point in proofreading the text until the key issues are covered:

By definition, DEF is the ratio of absorbed doses in a volume with and without nanoparticles. The device presented by the authors does not measure the absorbed dose in the volume where the nanoparticles are located. It is necessary to indicate how the device readings (current, charge) and the absorbed dose in the volume under study relate.

What is required is to describe the physical meaning of the REF, not just the definition. For example, DEF potentially tells how the biological effect associated with the absorbed dose will change.

Data in Fig. 5 and Table 2 clearly show that the sensitivity of the device depends on the spectral composition of the radiation, which changes when passing through a substance depending on its atomic composition and size. This may be the basis for the observed dependencies.

Data in Fig. 6. indicate that the DEF value reaches a plateau with increasing concentration. In fact, this suggests that the absorbed dose does not depend on the elemental composition of the medium, which is fundamentally incorrect.

In Fig. 5 the data are shown on a logarithmic scale, i.e. linearity y=a*x+b is observed relative to the variables y=Ln(Q/Q0) and x=Ln(K/K0), where Q is charge, K is kerma. Potentiating this expression, we obtain that Q=Const*K^a. Those. nonlinear dependence.

Reviewer 4 Report

Comments and Suggestions for Authors

Authors addressed all the points raised up in my first round of revision 

Author Response

Thank you for your acceptance!